# Response of Bridge Foundation with Drainage Structure in the Liquefied Inclined Site under Sinusoidal Waves

**Zhixiong Chen [1], Buxin Wang [1], Xuecheng Gao [2,*] and Haocheng Yan [1]**

[1] College of Civil Engineering, Chongqing University, Chongqing 400045, China
[2] College of Environment and Ecology, Chongqing University, Chongqing 400030, China
* Correspondence: xuechengg@cqu.edu.cn

**Abstract:** Many earthquake damage investigations have shown that lateral spreading is one of the main causes of damage to bridge foundations. However, the seismic research on bridge foundations with drainage systems is relatively lacking. Therefore, based on the shaking table test, the seismic response of a drained sheet pile-reinforced bridge foundation on a liquefied inclined site was studied under the action of sinusoidal waves. Compared with the conventional group, the peak excess pore water pressure ratio and the lateral displacement of the sheet-pile wall of the test group were smaller, but the acceleration amplification factor was larger, indicating that the anti-liquefaction performance of the site was effectively improved. Meanwhile, the acceleration amplification factor of the test group was larger, and the lateral displacement of the bridge superstructure was smaller. These results indicated that the drainage structure significantly improved the stability and safety of the bridge system.

**Keywords:** lateral spreading; shaking table test; drainage sheet pile; bridge foundation





## 1. Introduction

Lateral spreading of saturated sand is a common seismic damage phenomenon. A large number of earthquake disaster studies have shown that liquefaction lateral spreading is the main cause of structural earthquake damage [1–6]. For example, Hamada et al. [7] proposed that the main reason for the collapse of Showa Bridge in the 1964 Niigata earthquake was that the pile group foundation in the liquefaction lateral spreading site had excessive horizontal displacement at the top of the pile. Zhou et al. [8] found that in the 2008 Wenchuan earthquake, the horizontal motion load of the bridge increased due to the liquefaction and expansion of soil mass, which led to beam cracks and the separation of beams and columns of Baihua Bridge. Werner et al. [9] reported lateral spreading around the foundation of a bridge near Port au Prince in the 2010 Haiti earthquake which led to horizontal movement of the bridge's pile foundation. The above studies on earthquake disasters have shown that liquefaction-induced lateral spreading can severely damage bridge foundations. Therefore, the question of how to reduce the damage caused by liquefaction-induced lateral spreading on bridge pile foundations as a result of earthquakes has become a scientific problem of concern in bridge engineering.

The liquefaction-induced lateral spreading of foundation soil is mainly caused by excess pore water pressure in the soil that cannot be reduced quickly enough. Therefore, scholars have conducted many experimental studies with the aim of preventing liquefaction by reducing pore water pressure. Tanaka et al. [10,11] studied whether a rigid-drainage pile with a porous channel was effective at preventing sand liquefaction and reducing the loss of soil strength. The results showed that a rigid-drainage pile could effectively reduce the settlement of the embankment and the lateral displacement of soil when used with a cofferdam embankment. Harada et al. [12] proposed a new drainage pipe design composed of multilayer horizontal steel rings and vertical drainage pipes which could

reduce the excess pore water pressure in soil during earthquake by collecting water in the steel rings and draining it through the steel pipes. Liu et al. [13] proposed a new drainage pile design that fixes the drainage structure to the pile body. Their research showed that the new drainage pile could dissipate excess pore water pressure in time through the drainage channel and reduce the degree of liquefaction at the site. Rasouli et al. [14] studied the influence of different groundwater levels on the structures of buildings equipped with multidirectional drainage pipes and sheet-pile walls. The results showed that the liquefaction resistance of foundations that had drainage pipes was significantly higher than that of conventional foundations.

The above research mainly focused on seismic research into bridge pile foundations, but research on composite systems similar to superstructure sheet-pile walls—that is, pile group systems in liquefied lateral spreading sites—requires further study. To date, scholars have studied the effects of earthquakes on structures using various methods, such as shaking table tests [15–17], FEM (finite element method) analyses [18–21], etc. Following previous studies' methodology [22–28], a series of shaking table tests was conducted and the seismic response of a bridge foundation strengthened with drainage sheet piles in liquefied inclined sites was investigated. Various parameters of the response of laterally spreading foundations, such as excess pore water pressure ratio and acceleration, were tested. Parameters associated with the sheet-pile wall, including bending moments and lateral displacement, were also tested, and the lateral displacement of the bridge superstructure was recorded. The results are analyzed and discussed below. Based on the reported results, it can be concluded that bridge pile foundations with drainage structures are more stable in sites where liquefaction-induced lateral spreading occurs. Future areas of research on this topic are also proposed.

## 2. Test Setup and Model Preparation

### 2.1. Shaking Table Facility

The test was performed in the shaking table laboratory of the geotechnical experiment building of Chongqing University, China. The size of the shaking table was 1.2 m × 1.2 m (length × width), and the maximum horizontal acceleration was 2.0 g. The specific parameters of the shaking table are shown in Table 1. Based on previous research [29], a laminar shear box was used to reduce the boundary effect in this experiment. The internal dimensions of the laminar shear box were 950 mm × 828 mm × 650 mm (length × width × height).

**Table 1.** Parameters of the shaking table system.

| Parameters | Figure |
|---|---|
| Platform size | 1.2 m × 1.2 m |
| Maximum load | 1000 kg |
| Maximum horizontal acceleration | 2.0 g |
| Frequency range | 0~50 Hz |
| Maximum horizontal speed | 0.5 m/s |
| Maximum horizontal displacement | 100 mm |

In 1 g shake table tests on soils, the stress and strain conditions at scale are not representative of stress and strain conditions in the field. Therefore, based on previous studies [30,31], the Buckingham $\pi$ theorem and the similitude laws recommended by Iai [32] were employed in this study to design the scale model. Based on the size of the shaking table, the geometric similarity ratio was set to $S_L = 0.05$. The specific dimensions of the bridge foundation after reduction by the similar ratio is shown in Figure 1. Since the test was conducted in an environment with a gravitational acceleration of 1 g, the acceleration similarity ratio was $S_a = 1.0$, and the sand (coral sand) used in actual engineering was used in this test, so the density similarity ratio was $S_\rho = 1.0$. The similarity ratio of other parameters could be calculated according to the Buckingham $\pi$ theorem, as shown in Table 2.

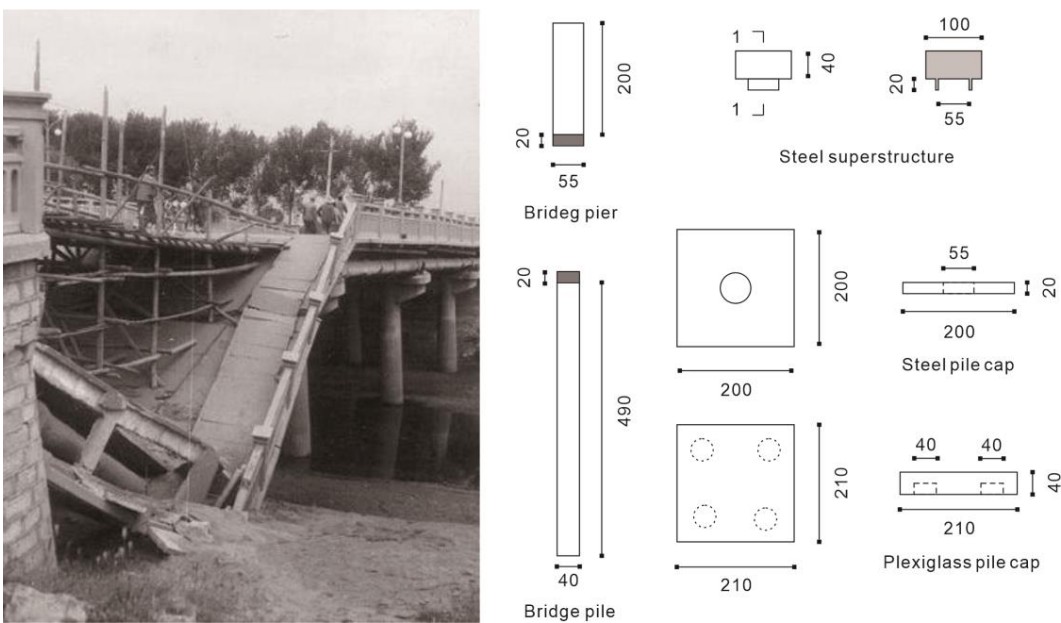

**Figure 1.** Victory Bridge after the Tangshan earthquake (unit: mm).

**Table 2.** Similitude ratios.

| Parameters | Physical Symbols | Conversion Formula | Similarity Ratio |
|---|---|---|---|
| Length | L | $S_{\mathrm{L}}$ | 0.05 |
| Density | ρ | $S_{\rho}$ | 1.0 |
| Poisson's ratio | μ | $S_{\mu}$ | 0.05 |
| Elasticity modulus | E | $S_{\mathrm{E}}$ | 0.05 |
| Acceleration | a | $S_a = S_{\mathrm{E}} S_{\mathrm{L}}^{-1} S_{\rho}^{-1}$ | 1.0 |
| Frequency | f | $S_f = S_{\mathrm{E}}^{0.5} S_{\mathrm{L}}^{-1} S_{\rho}^{-0.5}$ | 4.472 |
| Quality | m | $S_m = S_{\mathrm{L}}^{3} S_{\rho}$ | $1.25 \times 10^{-4}$ |
| Strength | F | $S_F = S_{\mathrm{L}}^{2} S_{\rho}$ | $1.25 \times 10^{-4}$ |
| Normal stress | σ | $S_{\sigma} = S_{\mathrm{E}}$ | 0.05 |
| Time | T | $S_T = S_{\mathrm{L}} S_{\mathrm{E}}^{-0.5} S_{\rho}^{0.5}$ | 0.224 |
| Cross-section | A | $S_A = S_{\mathrm{L}}^{2}$ | $2.5 \times 10^{-3}$ |
| Inertia moment | I | $S_{\mathrm{I}} = S_{\mathrm{L}}^{4}$ | $6.25 \times 10^{-6}$ |

### 2.2. The Structural Design of the Bridge

The prototype structure was Shengli Bridge, which collapsed in the Tangshan earthquake in 1976. Figure 1 shows the specific design structure, and the physical picture is shown in Figure 2.

The bridge foundation model used in this test was small, so it would have been difficult to control its quality had it been made using poured concrete. Therefore, this test used plexiglass and steel as counterweights to make the main body of the bridge foundation structure. The superstructure of the bridge foundation was made of steel and designed with reference to Shengli Bridge in Tangshan. The pile cap was divided into two layers: the upper pile cap was made of steel, and its length, width, and height were 200 mm, 200 mm, and 20 mm, respectively. The lower pile cap was made of plexiglass with a length, width, and height of 210 mm, 210 mm, and 40 mm, respectively. The drainage system was based on the rigid-drainage piles proposed by Liu et al. [13]. The drainage structure of the bridge foundation was wrapped with a thin layer of wire mesh and covered with a layer of geotextile, and then pasted on both sides of the pile. Because the drainage structure of the sheet-pile wall took into account the influence of the wall's thickness and strain gauge, the drainage structure was covered with a layer of geotextile, and then pasted on one side of the sheet-pile wall. The design of the sheet-pile wall was based on research by Partha et al. [33].

A plexiglass board with dimensions 410 mm × 580 mm × 6 mm (length × width × height) was adopted.

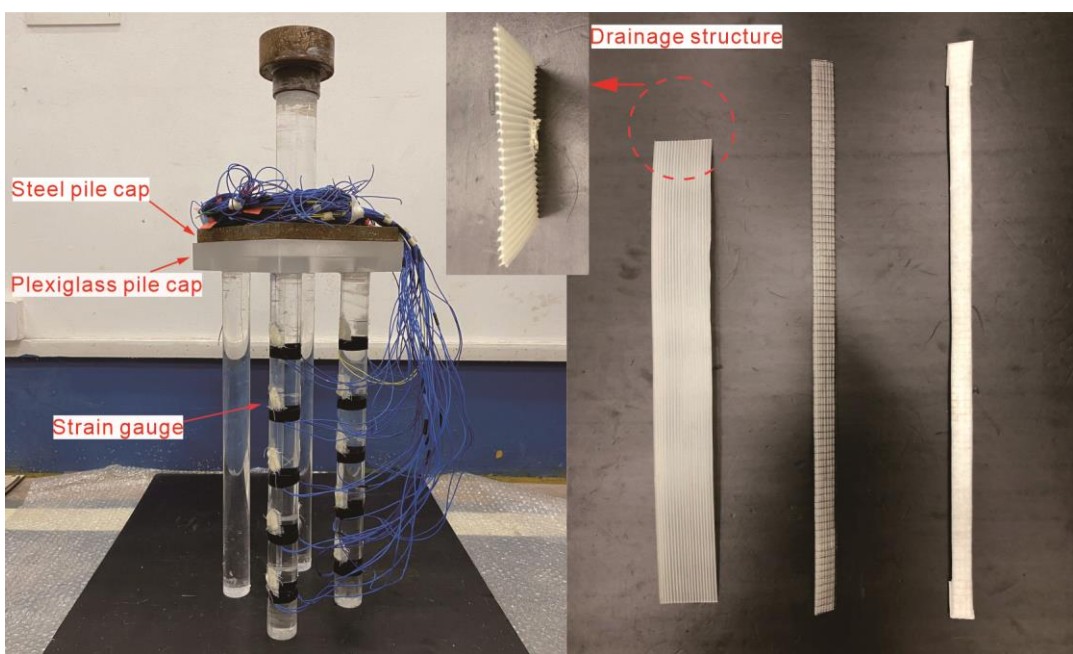

**Figure 2.** Annotated photographs of bridge foundation model.

### 2.3. Foundation Design

In this test, the pluviation deposition method was used to prepare the foundation soil. Coral sand was chosen as the test sand; its maximum dry density and minimum dry density were 1.48 g/cm³ and 1.15 g/cm³, respectively. Other important physical parameters of the coral sand are listed in Table 3.

**Table 3.** Physical properties of the sand.

| Type of Sand | $D_{50}$/mm | $C_u$ | $G_S$ | $\rho_d$, Max /(g/cm³) | $\rho_d$, Min /(g/cm³) |
|---|---|---|---|---|---|
| Coral sand | 0.48 | 2.67 | 2.8 | 1.48 | 1.15 |

The model foundation was prepared as follows. The laminar shear box was divided into two parts by an 8 mm thick plank along the direction of vibration to prevent the sites from interfering with each other. The left part was used to prepare the foundation for the test group (Case 1) and the right part was used to prepare the foundation for the conventional group (Case 2). Then the bridge and sheet-pile wall structures were fixed in the model box, as shown in Figure 3a. Next, the laminar shear box was filled with water at a depth of 6 cm each time, and then sand was sprinkled evenly into the laminar shear box by limiting the falling distance until the water surface was submerged; this operation was repeated. Accelerometers and pore water pressure transducers were placed at the heights specified in the design. In addition, in the test group, a layer of pebbles with a grain size of about 9 mm was placed around the pile cap, which was connected to the path of the drainage piles to form the drainage channel. Then an iron bracket was installed on the shaking table to fix the displacement transducers. Finally, the prepared soil foundation was left for 12 h to allow the foundation to become compact. Figure 3b shows the model foundation after the preparation process was completed.

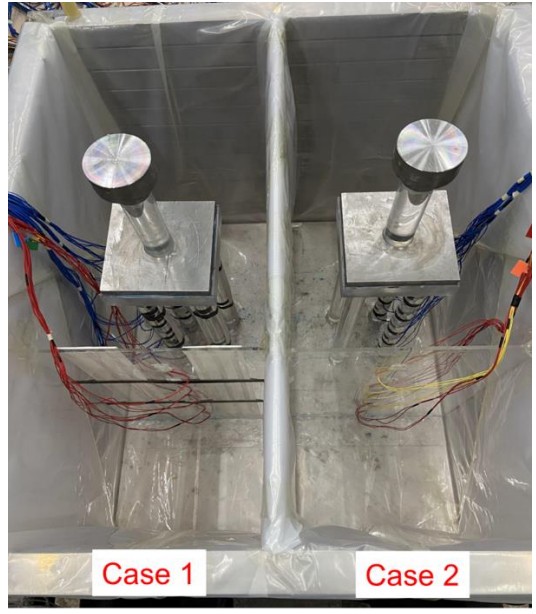
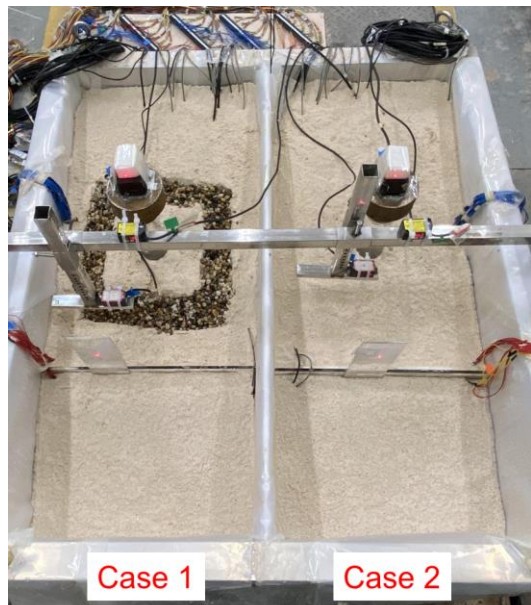

(**a**) Before preparing the model foundation      (**b**) After preparing the model foundation

**Figure 3.** Annotated photograph of the model foundation.

### 2.4. Layout of Components

For the model foundation, the layout of components were the same for the test group and the conventional group. Figure 4 shows the plan and schematic cross-section of the model foundation along with the layout of components. Strain gauges were symmetrically attached to model piles and the sheet-pile wall to measure the bending moment. To investigate the seismic response of the sand at different depths, both accelerometers and pore water pressure transducers were arranged in three layers. In addition, displacement transducers were fixed on the shaking table via an iron bracket.

### 2.5. Test Conditions and Loading Scheme

The test conditions are shown in Table 4. Since variation in sinusoidal waves is simpler than in other types of seismic wave, they are more conducive to analysis of acceleration, excess pore water pressure ratio, and other responses of objects in experimental conditions. Therefore, a large number of scholars choose to input sinusoidal waves in shaking table experiments [34–36]. In this paper, the experimental results under the action of sinusoidal waves were analyzed. Sinusoidal wave excitations with seismic intensities of PGA (peak ground acceleration) = 0.05 g, PGA = 0.10 g, and PGA = 0.20 g were input, and the time-history curves of the input acceleration measured. The results are shown in Figure 5. It should be noted that before each change in sinusoidal wave frequency, white noise scanning was adopted to obtain the natural frequencies of the model foundation. The relative density of the coral sand was 70% before the test. The loading sequence for each seismic level is shown in Table 5.

**Table 4.** Test conditions.

| Case | Bridge Pile Foundation | Sheet-Pile Wall | Type of Sand | Relative Density |
|------|------------------------|-----------------|--------------|------------------|
| 1 | Drainage | Drainage | Coral sand | 70% |
| 2 | Conventional | Conventional | Coral sand | 70% |

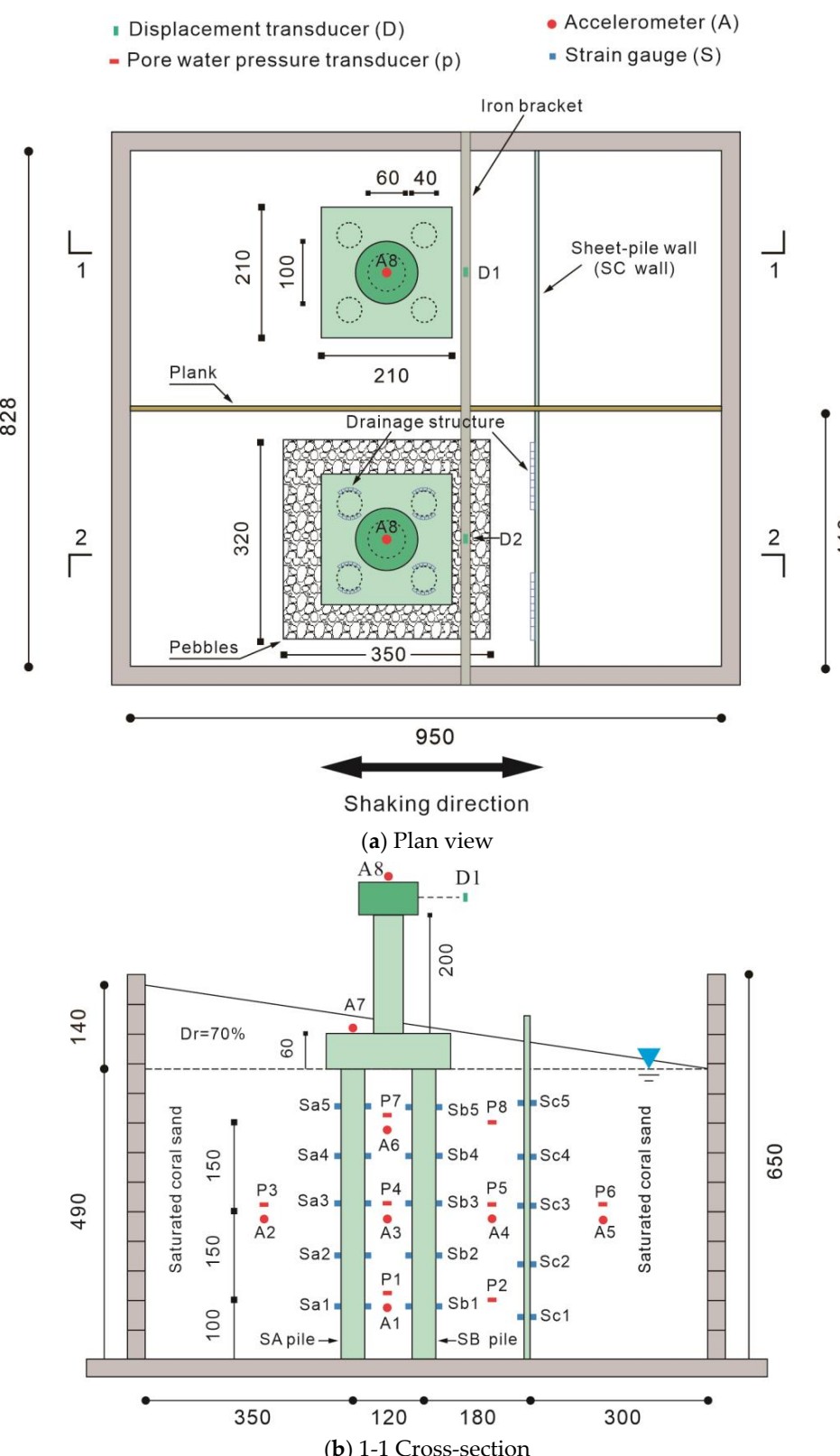

(**a**) Plan view

(**b**) 1-1 Cross-section

**Figure 4.** *Cont*.

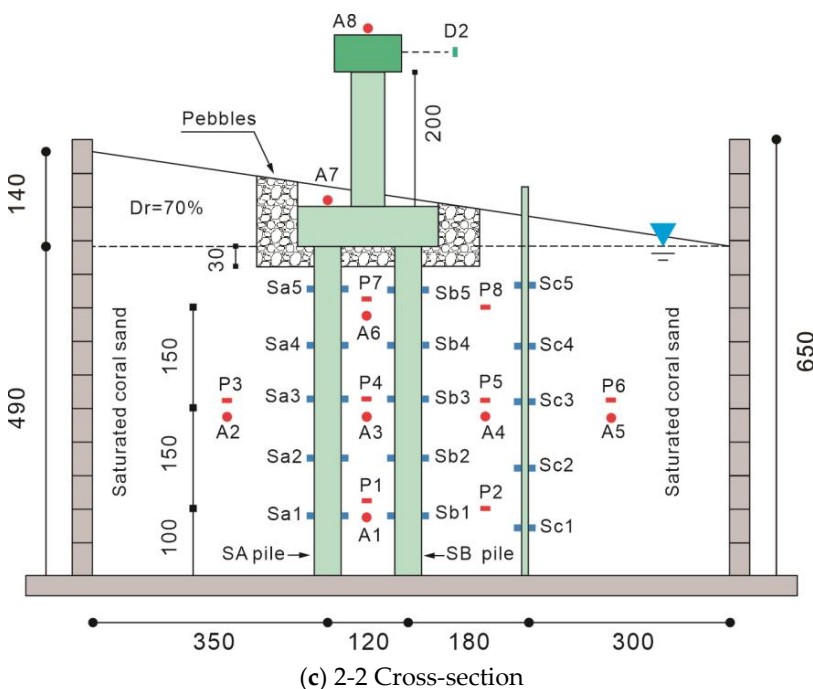

(**c**) 2-2 Cross-section

**Figure 4.** Component layout diagram (unit: mm).

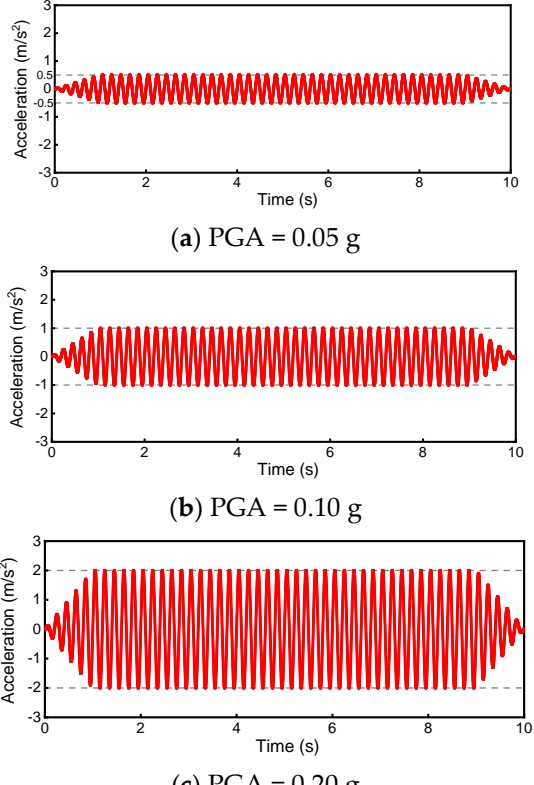

(**a**) PGA = 0.05 g

(**b**) PGA = 0.10 g

(**c**) PGA = 0.20 g

**Figure 5.** Time-history curves of the input acceleration.

**Table 5.** Loading sequence and test programs of the model test.

| Loading Sequence | Wave Form | Frequency (Hz) | PGA (g) | Duration (s) | Test Programs |
|---|---|---|---|---|---|
| 1 | White nose | | | | 1. Excess pore water pressure ratio. |
| | Sine wave | 5 | 0.05 | 10 | 2. Time-history curves of the acceleration. |
| 2 | White nose | | | | 3. Acceleration amplification factors. |
| | Sine wave | 5 | 0.10 | 10 | 4. Bending moment of the sheet-pile wall. 5. Lateral displacement of the bridge superstructure. |
| 3 | White nose | | | | 6. Lateral displacement of the sheet-pile wall. |
| | Sine wave | 5 | 0.20 | 10 | |

## 3. Experimental Results and Analysis

### 3.1. Excess Pore Water Pressure

To clearly describe the variations of excess pore water pressure during the vibration excitation, the excess pore water pressure ratio, $R_u$, was defined as follows:

$$R_u = \Delta u \, / \, \sigma \qquad (1)$$

where $\Delta u$ and $\sigma$ were the measured excess pore water pressure and the total stress, respectively. In previous studies, the excess pore water pressure ratio has often been defined as the ratio of excess pore water pressure to effective stress, with 1.0 used as the liquefaction standard [37]. The total stress was generally larger than the effective stress in the liquefaction process. Therefore, in this test, the criterion for liquefaction to occur was that $R_u$ reached 0.8.

The layout diagram of the pore pressure gauge is shown in Figure 4. Due to the limitations in article length, for this paper, only the excess pore water pressure of P1, P3, P4, P5, and P7 measurement points were analyzed when PGA = 0.10 g and PGA = 0.20 g. Furthermore, the seismic response was more significant when PGA = 0.10 g and PGA = 0.20 g. Figure 6 shows the time-history curve of excess pore water pressure ratio. In the figure, 2.5 s was taken as the start time of the vibration excitation input. Under the action of PGA = 0.10 g, the excess pore water pressure ratio of the test group was significantly lower than that of the conventional group. This is due to the fact that the excess pore water in the conventional group could only be discharged from the soil through the irregular drainage channels formed by the soil during vibration excitation. The drainage channels formed between the drainage structure and pebbles in the test group had better drainage effects than those in the conventional group, thus effectively reducing the development of excess pore water pressure. In addition, the peak excess pore water pressure ratio of the conventional group reached 0.8 at the P7 measurement point, indicating that the soil in the conventional group had been liquefied. However, the peak excess pore water pressure ratio of the test group also occurred at the P7 measurement point, but its value was only 0.6, meaning it did not reach the liquefaction standard. When PGA = 0.20 g, after inputting vibration excitation, the excess pore water pressure ratio of both sites underwent rapid increase of about 1.5 s to reach the peak value. After reaching the peak, the excess pore water pressure ratio of the conventional group was maintained at a higher level compared to that of the test group, but the difference between them was not as large as when PGA = 0.10 g. This indicates that the effect of the drainage structure in the test group decreased as the vibration intensity increased. When the vibration excitation stopped, the excess pore water pressure ratios of the test and conventional groups started to decrease.

Figure 7 shows the vertical and horizontal variation of the peak excess pore water pressure ratio in the two sites. Under the action of PGA = 0.10 g, the peak excess pore water pressure ratios at the three measurement points were less in the test group than in the conventional group, being 0.69, 0.66, and 0.62 times greater than in the conventional group along the depth. Meanwhile, the peak excess pore water pressure ratio decreased with the increased of buried depth, which indicates that when the vibration excitation is input, the upper layer of the site liquefies first, and then gradually the lower layer. This

finding is consistent with the general law of foundation liquefaction during earthquakes. In addition, from the measurement point furthest away from the sheet-pile wall to the measurement point nearest it, the excess pore water pressure ratio of the test group was 0.74, 0.66, and 0.44 times that of the conventional group, respectively. This indicates that the drainage structure can significantly reduce the development of excess pore water pressure in the foundation under the action of earthquakes. When PGA = 0.20 g, the peak excess pore water pressure ratio in the two sites was close. However, at the P5 measurement point, the peak excess pore water pressure ratio of the test group was much smaller than that of the conventional group. This is because the P5 measurement point was close to the sheet-pile wall. During the vibration excitation process, the drainage structure on the sheet-pile wall and the drainage structure on the pile worked together to effectively drain the excess pore water out of the soil. These results show that the peak excess pore water pressure ratio of the test group was always smaller than that of the conventional group under both PGA = 0.10 g and PGA = 0.20 g, indicating that the drainage structure could be effective at improving the liquefaction resistance of the foundation.

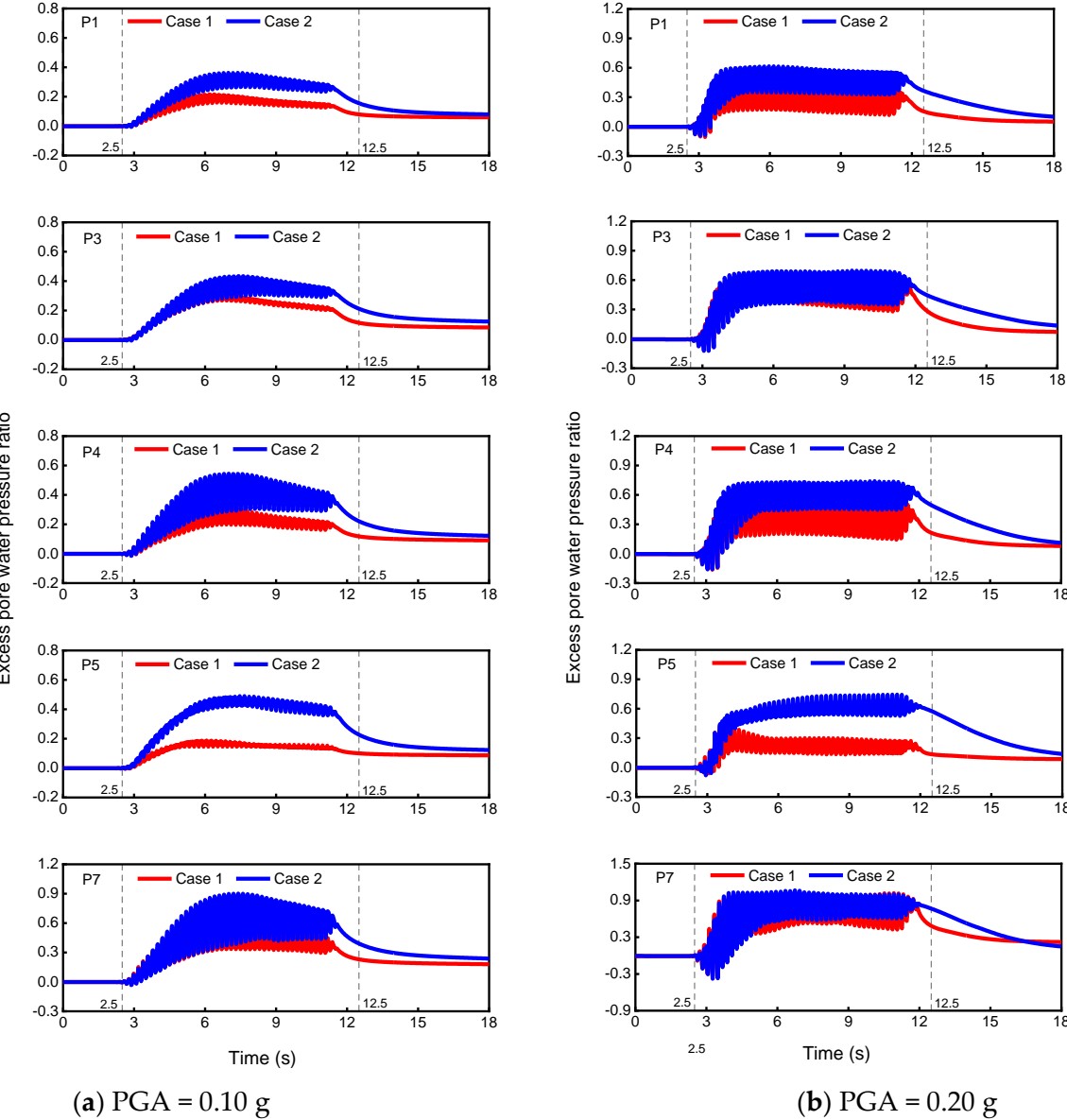

**(a)** PGA = 0.10 g

**(b)** PGA = 0.20 g

**Figure 6.** Time-history curve of excess pore water pressure ratio.

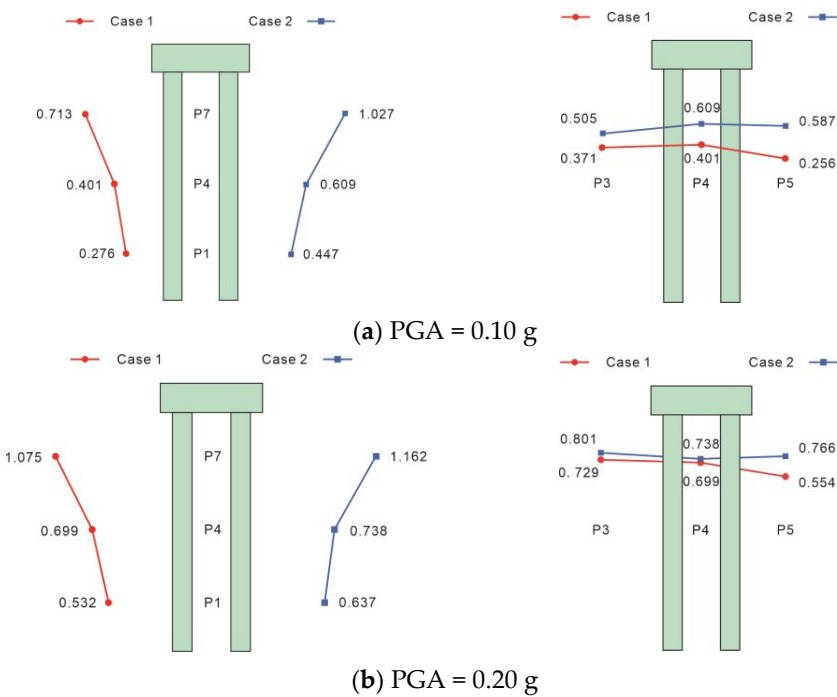

(**a**) PGA = 0.10 g

(**b**) PGA = 0.20 g

**Figure 7.** Distribution of the peak excess pore water pressure ratio along the vertical and horizontal directions.

*3.2. Acceleration*

This section compares and analyzes the time-history curves of the acceleration of the test group and the conventional group when PGA = 0.20 g. The time-history curves of the acceleration when PGA = 0.20 g are depicted in Figure 8; in the figure, 1.0 s was used as the start time of the vibration excitation input. To facilitate the analysis of the test results, the time-history curves were divided into three stages: Stage 1 (within 1.5 s after the vibration excitation input); Stage 2 (between 1.5 s and 2.0 s after the vibration excitation input); and Stage 3 (between 2.0 s and 10.0 s after the vibration excitation was input). In Stage 1, the acceleration of the conventional group increased rapidly and attained peak acceleration. At this time, the foundation was not yet liquefied and had a certain shear strength. In Stage 2, the acceleration of the conventional group rapidly decreased to about half the peak acceleration, indicating that the site liquefied and the soil lost most of its shear strength. In Stage 3, the acceleration of the conventional group remained almost constant. On the whole, in Stage 3, the peak acceleration of the test group was larger than that of the conventional group, indicating that the drainage structure could effectively maintain the shear strength of the foundation soil and had a reinforcement effect on the bridge pile foundation. It should be noted that the acceleration of the A7 measurement point in the conventional group gradually decreased in Stage 3 because the A7 measurement point was close to the ground surface and the liquefaction was intense. In addition, the time-history curves of the acceleration recorded at the A3 and A8 measurement points of the conventional group were asymmetric, which may have been caused by the tilting of the accelerometer during the vibration excitation.

Figure 9 shows the acceleration amplification factors (i.e., the ratio of peak acceleration at the measurement point to peak acceleration input from the shaking table) of measurement points A1, A2, A3, A4, and A6. Under the action of PGA = 0.10 g, the acceleration amplification factors of both sites were larger than 1, and gradually decreased with the increase of buried depth, indicating that the upper layer of the foundation would be liquefied first, which conforms to the general laws of foundation liquefaction during earthquakes. The acceleration amplification factors of the measurement points in the test group were 1.06, 1.04, and 1.03 times those of the conventional group from top to bottom,



and 1.05, 1.04, and 1.02 times those of the conventional group from far away from the sheet-pile wall to near the sheet-pile wall. When PGA = 0.20 g, the acceleration amplification factors of the two sites were greater than those measured under the action of PGA = 0.10 g, and the acceleration amplification factor of the test group was still larger than that of the conventional group, indicating that the drainage structure could discharge the excess pore water in the soil in a timely manner, thus maintaining a higher shear strength in the foundation.

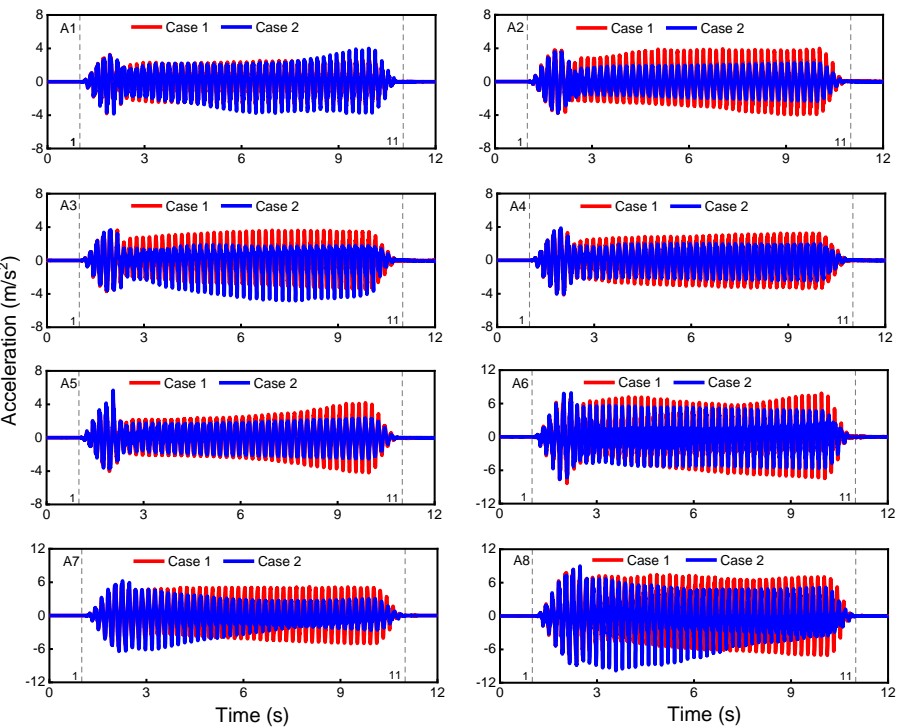

**Figure 8.** Time-history curves of the acceleration (when PGA = 0.20 g).

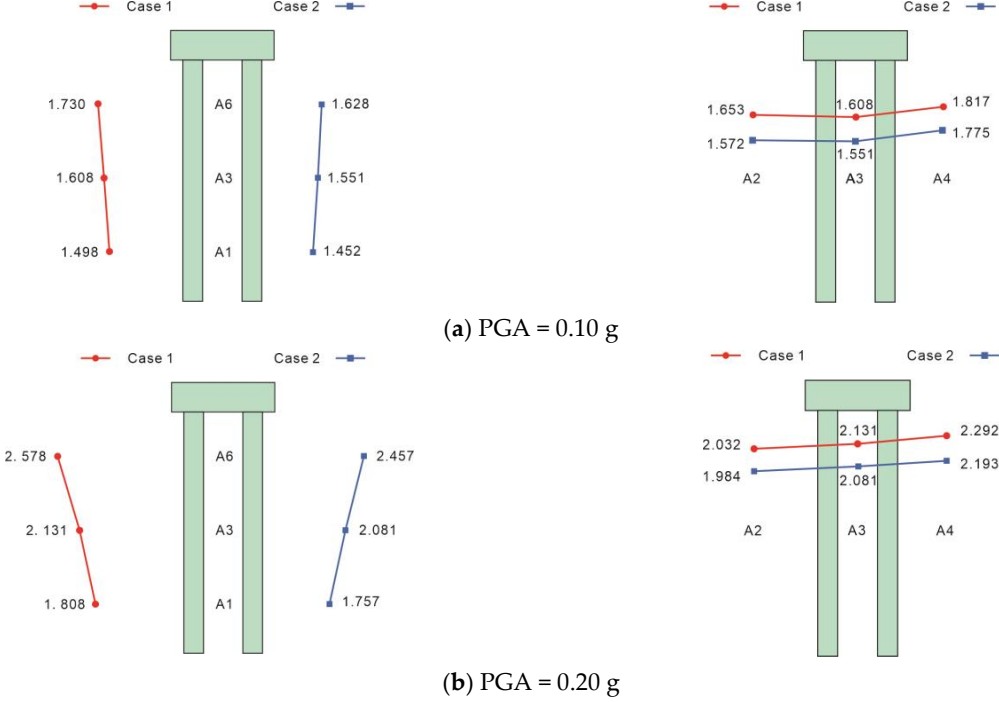

(**a**) PGA = 0.10 g

(**b**) PGA = 0.20 g

**Figure 9.** Distribution of the acceleration amplification factor along the vertical and horizontal directions.

### 3.3. Pile Bending Moment

In the test, strain gauges were pasted symmetrically on both sides of each pile and sheet-pile wall along the inclined direction of the site. The specific distribution is shown in Figure 4. According to the Euler–Bernoulli beam theory [38], the bending moment, *M*, can be obtained from the following form:

$$M = \frac{EI(\varepsilon_t - \varepsilon_c)}{D} \tag{2}$$

where *EI* (*E* = 2.8 GPa) is the bending stiffness of the pile or sheet-pile wall cross-section, $\varepsilon_t$ and $\varepsilon_c$ represent the tensile and compressive strains at a particular measurement point measured by strain gauges, and *D* is the diameter of the pile or the thickness of the sheet-pile wall. In this paper, the bridge piles furthest away from the sheet-pile wall were called SA piles, the bridge piles near the sheet-pile wall were called SB piles, and the sheet-pile wall was called the SC wall.

In previous experimental studies [39,40], only the bending moment on the pile body has been analyzed, with no consideration of the variation in bending moment in the sheet-pile wall. Therefore, this section will explore the distribution law of the bending moment on the sheet-pile wall.

Figure 10 shows the distribution of bending moments along the depth of the sheet-pile wall under the action of PGA = 0.10 g. Under the action of PGA = 0.10 g, the sheet-pile walls' maximum bending moments in the test and conventional groups were located in the middle (Sc3 measurement point) of sheet-pile walls, which were 1.663 N·m and 2.803 N·m, respectively, the latter being about 1.7 times greater than the former. The bending moment at the bottom (Sc1 measurement point) of the sheet-pile wall was larger than that at the top (Sc5 measurement point) in both sites, because the bottom of the sheet-pile wall was fixed with special glue and the bottom fixing of the sheet-pile wall could also withstand part of the bending moment. Overall, the bending moment of the sheet-pile wall in the test group was generally smaller than that in the conventional group, which indicates that the drainage structure on the sheet-pile wall in the test group could effectively drain the excess pore water and reduce the liquefaction of the soil around the sheet-pile wall. However, there was no drainage structure in the conventional group, so the excess pore water pressure around the sheet-pile wall could not be dissipated in time, resulting in a higher degree of soil liquefaction than that in the test group. In addition, the soil lost shear strength after liquefaction, producing more serious lateral spreading and greater lateral forces, which made the bending moment of the sheet-pile wall in the conventional group larger than that in the test group. The results indicate that sheet-pile walls reinforced by drainage structures are more stable during liquefaction.

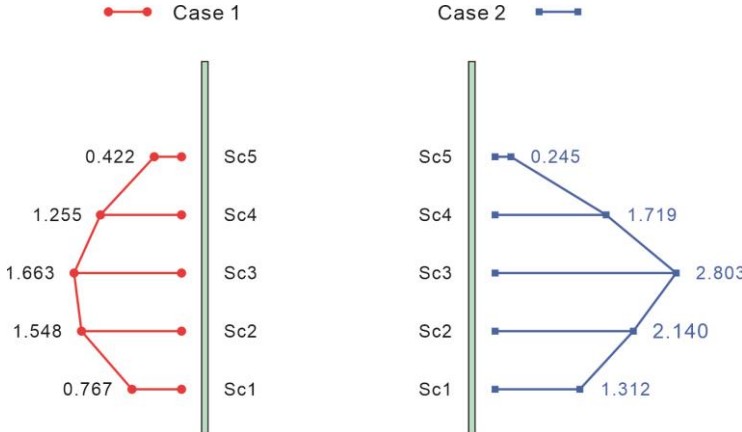

**Figure 10.** Distribution of the bending moment along the depth of sheet-pile walls (PGA = 0.10 g). (Unit: N·m).

### 3.4. Displacement

Time histories of lateral displacements of the bridge superstructure in two sites are depicted in Figure 11. Under the action of PGA = 0.10 g, the lateral displacement of the superstructure of the conventional group first experienced a slow rise, then reached a peak value about 5 s after the vibration excitation input, and remained stable until the vibration excitation input ended. The lateral displacement of the superstructure of the test group reached its peak value when the vibration excitation input was about 3 s. During the vibration excitation process, the vibration amplitude and residual displacement of the test group were smaller than those of the conventional group. After the vibration excitation, the residual displacements of the test and conventional groups were 2.07 mm and 2.59 mm, respectively. When PGA = 0.20 g, the lateral displacement of the superstructure of the test group developed rapidly and peaked within 1.5 s of the vibration excitation input, and then remained stable until the vibration excitation input stopped. However, the lateral displacement of the superstructure of the conventional group increased slowly throughout the vibration excitation input. Eventually, the residual displacement of the superstructure was 3.39 mm and 8.33 mm for the test and conventional groups, respectively. From the test results, it is known that when PGA = 0.10 g and 0.20 g, the residual displacement of the superstructure in the test group is smaller than that in the conventional group. The reason for this is that the drainage structure in the test group enabled the excess pore water to discharge in time, whereas in the conventional group it could not, resulting in a reduction in the effective stress of the soil and a reduction in shear strength, which eventually caused the bridge superstructure to produce greater lateral displacement.

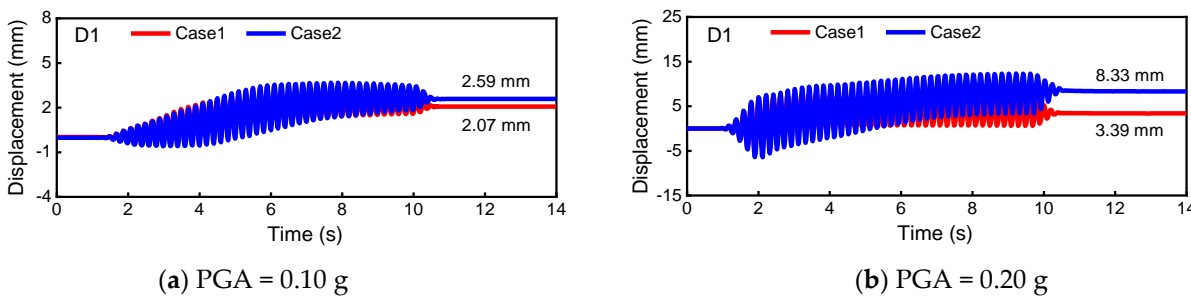

(**a**) PGA = 0.10 g                                            (**b**) PGA = 0.20 g

**Figure 11.** Lateral displacement of the bridge superstructure.

Liquefaction of soil leads to the loss some shear strength, and the lateral flow of liquefied soil also causes the lateral displacement of the sheet-pile wall, so the lateral displacement of the sheet-pile wall could also indicate the degree of liquefaction of the site. Figure 12 shows the lateral displacement of the sheet-pile wall after each round of vibration excitation input. When PGA = 0.05 g, the sheet-pile walls of the test group and the conventional group did not produce lateral displacement. With a PGA of 0.10 g, the lateral displacements of the sheet-pile wall in the conventional group and the test group were 14 mm and 8.5 mm, respectively, the former being significantly larger than the latter. This is due to the fact that in the process of vibration excitation input, the excess pore water in the conventional group could not be dissipated immediately, and the lateral flow of the liquefied soil increased the soil pressure on the sheet-pile wall, eventually leading to a greater lateral displacement of the sheet-pile wall. In contrast, the drainage structure of the test group dissipated some of the excess pore water, which better preserved the effective stress of the foundation soil and limited the lateral displacement of the sheet-pile wall. Under PGA = 0.20 g, the lateral displacement of the sheet-pile wall of the test group was still smaller than that of the conventional group, 19 mm and 19.5 mm, respectively. The reason for this is that the lateral displacement of the conventional group sheet-pile wall was greater than that of the test group under the moderate excitation load (PGA = 0.10 g), meaning that the compactness of the soil in the conventional group was greater and the soil was further stabilized after the moderate excitation load. Therefore, when PGA = 0.20 g,

the difference in lateral displacement between the two groups of sheet-pile walls was not significant.

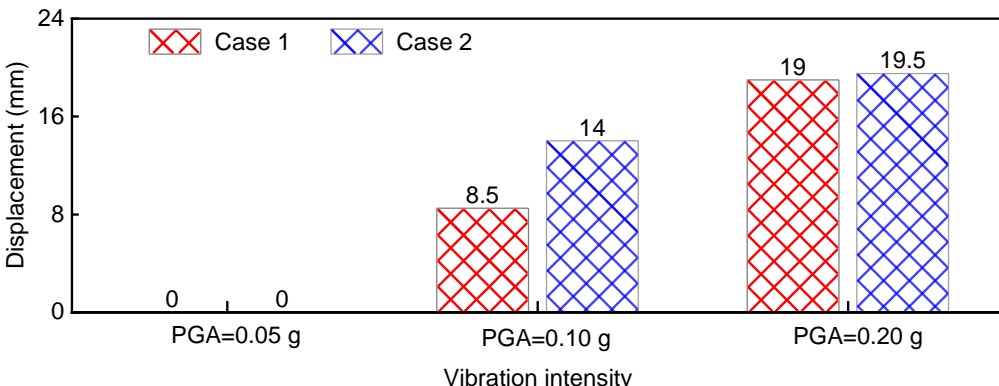

**Figure 12.** Lateral displacement of the sheet-pile wall.

## 4. Discussion

In this paper, the responses of conventional bridge foundations and those with with drainage structures to sinusoidal waves was investigated by means of shaking table tests. The excess pore water pressure ratio, time-history curves of acceleration, acceleration amplification factors, bending moment and lateral displacement of sheet-pile walls, and lateral displacement of bridge superstructures were compared and analyzed for a test group and a conventional group.

In previous studies on liquefaction-induced lateral spreading [40–43], scholars have generally used the results of shaking table tests to elucidate the behavior of pile foundations subjected to liquefaction-induced lateral spreading; however, the question of how to reduce the damage to bridge foundations and superstructures has not yet been considered. The highlight of this paper is the design of a comparative test to verify that bridge foundations with drainage structures are more stable than conventional bridge foundations in a liquefaction-induced lateral spreading site.

We found that as vibration excitation intensity increased, the differences in the excess pore water pressure ratio between the test and conventional groups were not as large overall as at PGA = 0.10 g when PGA = 0.20 g. Therefore, the increase in vibration excitation intensity reduced the function of the drainage structure. More shaking table tests with various scenarios should be conducted to investigate the liquefaction resistance of bridge pile foundations with drainage structures under the action of vibration excitation of different intensities. And the influence of soil structure and pore fluid on the development of excess pore water pressure was not taken into account during the test, further tests on this aspect can be carried out.

In addition, due to the limited test conditions in this study, as well as the influence of artificial factors in the production of the model foundation, among other factors, a large number of test results are needed to verify the dynamic response law of the bridge foundations with drainage structures in shaking table tests.

## 5. Conclusions

A series of shaking table tests was conducted to study the seismic response of a bridge foundation strengthened with drainage sheet piles in an inclined site experiencing coral sand liquefaction. The results show that the drainage structure had a protective effect on the bridge foundation during vibration excitation. This study can therefore act as a valuable reference for the design of bridge pile foundations in applied engineering. The primary conclusions of this study are as follows:

(1) Under the action of vibration excitation of different intensities, the peak excess pore water pressure ratio of the test group was smaller than that of the conventional group, indicating that the drainage structure could effectively reduce the excess pore water pressure ratio of foundation soil, thus reducing the possibility of liquefaction. As buried depth increased, the excess pore water pressure ratio became smaller. Therefore, in practical engineering applications, attention should be given to the treatment of the surface layer of foundation soil.

(2) The acceleration amplification factors of the two types of site decreased gradually with the increase of buried depth, indicating that the upper layer of the foundation would liquefy first, with the change gradually transferred downward, and that a drainage structure could effectively reduce the peak value of the acceleration response curve.

(3) The maximum bending moment of the sheet-pile walls occurred in their center, and the bending moment at the bottom was larger than that at the top. For the sheet-pile wall reinforced by the drainage structure, the bending moment of the test group was significantly reduced compared with the conventional group, indicating that the drainage structure could improve the stability of the sheet-pile wall during the liquefaction process.

(4) The drainage structure was able to effectively reduce the lateral displacement of the bridge superstructure. The effect was more obvious as the intensity of the input vibration excitation increased. The drainage structure was also able to reduce the lateral displacement of the sheet-pile wall. This shows that the drainage structure improved the liquefaction resistance of the site and the stability of the bridge system.

**Author Contributions:** Formal analysis, methodology, writing—review & editing, funding acquisition, Z.C.; writing—original draft, B.W.; funding acquisition, supervision, X.G.; investigation, testing, data curation, H.Y. All authors have read and agreed to the published version of the manuscript.

**Funding:** This research was funded by the National Natural Science Foundation of China (Grant No.52278331, 51908087, 51778092), the Postdoctoral innovative talents support program, Chongqing (Grant No. CQBX2021020), Chongqing.

**Institutional Review Board Statement:** Not applicable.

**Informed Consent Statement:** Not applicable.

**Data Availability Statement:** The data is available through the author.

**Acknowledgments:** This work was supported by the National Natural Science Foundation of China (Grant No.52278331, 51908087, 51778092), the Postdoctoral innovative talents support program, Chongqing (Grant No. CQBX2021020), Chongqing.

**Conflicts of Interest:** The authors declare no conflict of interest.

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
