# Peer review of "Response of Bridge Foundation with Drainage Structure in the Liquefied Inclined Site under Sinusoidal Waves"

_applsci, doi:10.3390/app13021009_

Round 1

Reviewer 1 Report

The article addresses an important and very interesting topic of the seismic response of bridge foundation with drainage sheet in liquefied inclined site, which is appreciated. The study includes the experimental research. In this paper, a series of shaking table tests were conducted, and the seismic response of the bridge foundation strengthened by drainage sheet piles in liquefied inclined sites was investigated. The Reviewer has some concerns regarding the introduction, description of research, results, discussion of results, conclusions and references. Generally, in this paper the English language should be checked by the Native Speaker. In opinion of Reviewer this paper should be subjected to major revision.

Other comments:

1.          The title of this paper should be change, because in this research was analysed “sinusoidal waves” – lines 166-167. Thus, please improve this title adequate to the research.

2.          General remarks to paper – which (or how) the seismic impact was analysed in this research? In my opinion this is very important question according to e.g. title of research, approach this research etc.

·       Below you can find example paper about seismic analysis in bridge/tunnel/piles:

ü  https://doi.org/10.1061/(ASCE)GM.1943-5622.0002378

ü  https://doi.org/10.1016/j.soildyn.2021.107008

ü  https://doi.org/10.1007/s10518-021-01244-4

ü  https://doi.org/10.1016/j.tust.2020.103808.

ü  https://doi.org/10.12989/scs.2022.42.6.747

ü  https://doi.org/10.3390/geosciences9110473

ü  https://doi.org/10.1007/s10518-016-9975-7

3.          Introduction.

·       In my opinion the introduction should be synthetic and concise part of the text. In my opinion you should improve this aspect about aspect from point 2 of this Review (seismic impact).

4.          Model scale

·       Please explain in more detail how the real model was scaled to laboratory dimensions. Lines 76 – 81 are not clear.

5.          Figures

·       In the Figure 1 the Reviewer find the typos.

·       In Figure 2 the Reviewer cannot find “a” and “b” - line 130.

·       Lines 149 – 150 – please show this description on the figure. In current version are not clear.

In addition, displacement transducers were fixed on the shaking table through an iron bracket.”

·       Figures 5-8 – please show maximal values on the figures.

·       Figures 12 is necessary? what does this picture show?

6.          Discussion

·       The Reviewer cannot see the discussion of results? Please compare your results from results other researches.

7.          Conclusion

·       This is end of your research? Please improved this point.

8.          References

·       This part should be improved. Generally, the scientific paper should be based on the literature from all world, thus please check the literature from the best Journals.

Finally, I hope that my comments will be helpful for Authors.

Reviewer 2 Report

The paper submitted by the authors entitled "Seismic response of bridge foundation with drainage sheet in liquefied inclined site" presents the results of a series of 1-g shaking table tests to investigate the response of the drained sheet pile reinforced bridge foundation in the liquefied inclined site under sinusoidal waves. This is an important topic that warrants research, and the authors are commended for taking on this difficult problem. I believe that the paper fits into the aims and scope of Applied Science, and the paper can be considered for publication in the journal if the following aspects of the manuscript are improved. Please consider my comments as friendly suggestions to improve the quality of the paper and good luck with your revision.

Main comments:

1. Although the paper is generally suitably written, its structure and English should be improved. There are also several grammatical errors to be corrected (i.e., line 37).

2. Mitigation of liquefaction effects (i.e., lateral spreading) on the bridge foundations is of engineering importance. The attempt of this study is very good and appreciated, but the methodology (1-g shaking table testing) used by the authors appears doubtful. In 1-g shake table tests on soils, the stress and strain conditions at scale are not representative of stress and strain conditions in the field. In order to overcome this issue, researchers usually use scaled-down models (i.e., centrifuge models) and use similitude laws to scale for the most desirable variables. Without this, the results are not representative of field conditions. The authors should clearly indicate this limitation in the manuscript.

3. The authors report that using a bridge foundation with a drainage sheet in liquefied inclined site reduces the generation of excess pore pressure ratio and lateral displacement of the sheet-pile wall, but amplifies the accelerations and seismic forces transferred to the structures. Liquefaction mitigation is an expensive item that owners are reluctant to pay for unless it is necessary and if potential damage of mitigation cannot be tolerated most owners would choose other options. Therefore, further discussion on the negative effects of this method (i.e., increased acceleration) would be useful for both researchers and practitioners working in this field.

Specific comments:

Lines 8-14: The first four sentences of the abstract present basic knowledge and background information on bridge foundation, which is apparently more than necessary. Please be concise and avoid using language that is vague and exaggerated. For instance, you can simply use “important” instead of “more and more important”. You can also use phrases like “mitigation of liquefaction-induced lateral spreading damages on the bridge pile foundation”.

Lines 37-41: Please revise these sentences in 34-41. There are grammatical errors. The same applies to sentences lines 42-43. Also please be concise throughout the manuscript. Some sections are excessively wordy.

Lines 61-62: “Shaking table model could accurately reproduce seismic phenomena [15-21]” seems to be misleading. Please indicate the limitation of 1-g shaking table testing, as mentioned above.

Lines 71-72: Boundary effects (i.e., stress wave reflections in the direction of earthquake loading and lateral constraint on the monotonic movement of soil due to failure mechanisms) are important considerations in physical modeling tests. Incident stress waves reflected back into the soil can cause significant problems. The design of a laminar box whose model boundaries match the dynamic stiffness of the soil is important. The authors did not provide any information on the design of the laminar box they used. The authors are suggested to either give a reference or provide brief information on the model box. The same applies to the 1-g shaking table apparatus.

Lines 104-107: Please be consistent with the text font size. Moreover, please reconsider re-writing sentences in lines 104-106.

Line 119: Please be specific with the pluviation method you used. Is it wet pluviation? You can also refer to the corresponding standards you used for emax-emin tests.

Line 125: The authors separated the laminar box into two sides, which may increase the possibility of boundary effects. How do authors ensure that there are no boundary effects? Can an 8 mm thick plank simulate zero lateral stiffness?

Lines 130-131: It appears that the authors used water as the pore fluid. Do the authors think that using water (rather than viscous pore fluid) has some effects on the generation and dissipation of excess pore pressures? It is useful to add some discussion on this aspect.

Lines 134-136: Please reconsider re-writing sentences in lines 134-136. It is difficult to read.

Lines 136-137: Further details on the model preparation (i.e., size of pebbles) would be useful for the repeatability of the tests.

Lines 139-140: Left for 12 hours... Why 12 hours?

Lines 159-168: Please add a table that gives a summary of the test program. How many tests are conducted and what are the test conditions? It is difficult to follow. Table 4 is not sufficient for this. You can give an ID for each test and provide details for each test (wave type, relative density seismic motion characteristics). In this table, you can also include a summary of test results (i.e., excess pore pressure ratios, and structural displacements).

Lines 159-172: The authors used sinusoidal waves, and explained the rationale behind their selection. The sentences in lines159-166 should be shortened and restructured. It is confusing in its form. To me, it sounds like the authors used different wave types to compare, but this is not the case. The whole section between lines159-172 should be also rewritten for the sake of clarity.

Line 174: The input motions in Figure 5 look strange, starting with small cycles and then increasing in magnitude. It is useful to point out the start and end of the earthquake in the figure. The authors mention in line 193 that the start of the earthquake is 2.5 s but please display this in the figure.

Lines 182-184: The authors indicate that the excess pore water pressure ratio being equal to 0.8 is defined as liquefaction, but this appears to be inconsistent with many studies in the field of liquefaction. The ru=1 criterion is usually used as the onset of liquefaction (initial liquefaction), which is particularly true for one-directional element testing (i.e., simple shear test). It s true that actual earthquake events produce multi-directional loading, during which excessive strains can develop even for lower ru values (i.e., 0.8), and the resistance of soils during multi-directional tests is consequently lower than that during one-directional tests. In this case, ru value lower than 1 (such as 0.8) may mark the point above which softening is intensified, and the onset of softening allows increased strains. It would be useful for the authors to (more clearly) highlight why they chose ru=0.8 as the liquefaction criteria. The authors’ explanation seems to be insufficient in this regard.

Lines 178-354: The findings presented in the results section (particularly excess pore pressure behavior) appear to be the repetition of the published literature. If available, the authors are recommended to add more data, which could have increased the value of this paper (Optional). Alternatively, clearly highlight, where appropriate, the limitations of the study and practical usage of the findings presented in this work. Moreover, clearly highlight its novelty wherever appropriate.

Lines 178-354: Excess pore pressures start to (slightly) dissipate in some cases regardless of the presence of the drainage structures. Why is that? Is it to do with the frequency and magnitude of the earthquake loading? If this is the case, the authors would have used much larger frequencies. Another possibility might be partial drainage. Please add some discussion on this aspect if necessary.

Lines 178-354: Excess pore pressure ratios being more than 1 also needs explanation. The development of excess pore pressure at the sheet pile-soil interface is much higher or larger than that in the free field (away from the structures). Did the authors consider the soil structure effects?

Lines 367-369: Please clearly highlight, where appropriate, the limitations of the study and practical usage of the findings presented in this work. Moreover, it is recommended to add some tangible evaluations to provide more concrete conclusions to practicing engineers regarding this specific liquefaction mitigation method.

Round 2

Reviewer 1 Report

Thank you for your improving. The Reviewer has some concerns regarding the introduction and references. Introduction text should be based on literature from whole World in current version the Reviewer cannot see. In addition, please add a few lines about your next research.
